# Analysis of chemical exchange in iridium N-heterocyclic carbene complexes using heteronuclear parahydrogen-enhanced NMR
Charbel D. Assaf[1] ✉, Xin Gui [2], Oleg G. Salnikov [3], Arne Brahms [4], Nikita V. Chukanov[3], Ivan V. Skovpin[3], Eduard Y. Chekmenev [5], Rainer Herges [4], Simon B. Duckett [6], Igor V. Koptyug [3], Kai Buckenmaier[7], Rainer Körber[8], Markus Plaumann[9], Alexander A. Auer[2], Jan-Bernd Hövener [1] & Andrey N. Pravdivtsev [1] ✉

The signal amplification by reversible exchange process (SABRE) enhances NMR signals by unlocking hidden polarization in parahydrogen through interactions with to-be-hyperpolarized substrate molecules when both are transiently bound to an Ir-based organometallic catalyst. Recent efforts focus on optimizing polarization transfer from parahydrogen-derived hydride ligands to the substrate in SABRE. However, this requires quantitative information on ligand exchange rates, which common NMR techniques struggle to provide. Here, we introduce an experimental spin order transfer sequence, with readout occurring at $^{15}N$ nuclei directly interacting with the catalyst. Enhanced $^{15}N$ NMR signals overcome sensitivity challenges, encoding substrate dissociation rates. This methodology enables robust data fitting to ligand exchange models, yielding substrate dissociation rate constants with higher precision than classical 1D and 2D $^1H$ NMR approaches. This refinement improves the accuracy of key activation enthalpy $\Delta H^{\ddagger}$ and entropy $\Delta S^{\ddagger}$ estimates. Furthermore, the higher chemical shift dispersion provided by enhanced $^{15}N$ NMR reveals the kinetics of substrate dissociation for acetonitrile and metronidazole, previously inaccessible via $^1H$ NMR due to small chemical shift differences between free and Ir-bound substrates. The presented approach can be successfully applied not only to isotopically enriched substrates but also to compounds with natural abundance of the to-be-hyperpolarized heteronuclei.

Hyperpolarization techniques have emerged as powerful tools in nuclear magnetic resonance (NMR). They enable the amplification of NMR signals by several orders of magnitude beyond those achievable under thermal equilibrium conditions[1]. Despite NMR widespread use, its improvement has been necessary because NMR suffers from inherently low sensitivity.

Dissolution dynamic nuclear polarization (dDNP) is the leading technique to polarize liquids for in vivo MRI diagnosis[2]. However, this method's high cost and low throughput are considered significant drawbacks.

Among other hyperpolarization techniques, signal amplification by reversible exchange (SABRE) has garnered significant attention due to its

[1]Section Biomedical Imaging, Molecular Imaging North Competence Center (MOIN CC), Department of Radiology and Neuroradiology, University Medical Center Kiel, Kiel University, Am Botanischen Garten 14, 24118 Kiel, Germany. [2]Max-Planck-Institut für Kohlenforschung, Kaiser-Wilhelm-Platz 1, 45470 Mülheim an der Ruhr, Germany. [3]International Tomography Center SB RAS, 3A Institutskaya St., 630090 Novosibirsk, Russia. [4]Otto Diels Institute for Organic Chemistry, Kiel University, Otto- Hahn Platz 4, 24118 Kiel, Germany. [5]Department of Chemistry, Integrative Biosciences (Ibio), Karmanos Cancer Institute (KCI), Wayne State University, Detroit, MI, 48202, USA. [6]Centre for Hyperpolarization in Magnetic Resonance (CHyM), University of York, Heslington, YO10 5NY, UK. [7]High-Field Magnetic Resonance Center, Max Planck Institute for Biological Cybernetics, Max-Planck-Ring 11, 72076 Tübingen, Germany. [8]Physikalisch-Technische Bundesanstalt (PTB), Abbestraße 2-12, 10587 Berlin, Germany. [9]Otto-von-Guericke-University Magdeburg, Institute for Molecular Biology and Medicinal Chemistry, Leipziger Str. 44, 39120 Magdeburg, Germany. ✉e-mail: charbel.assaf@rad.uni-kiel.de; andrey.pravdivtsev@rad.uni-kiel.de

versatility and efficiency[3–9]. SABRE enables the utilization of hyperpolarized substrates while preserving their chemical composition. It achieves this by temporarily binding parahydrogen (pH$_2$) and a target to an Ir-catalyst. Such spin order is transferred through the resulting scalar coupling network[10–14]. Subsequent dissociation facilitates the accumulation of polarized substrate in solution (Fig. 1). This process can be used to rapidly analyze biofluids with a micromolar accuracy[15,16] or subsequent in vivo imaging[4,5]. SABRE has made significant advances during the last few years, enabling the large-scale polarization of pyruvate and its derivatives to more than 10% $^{13}$C polarization values[4,17]. Unlike dDNP, pH$_2$-based techniques are scalable and offer a short duty cycle. The pursuit of hyperpolarization techniques represents a compelling avenue for pushing the boundaries of NMR and expanding its utility in various medical, scientific, and industrial applications.

From the very first description of the SABRE effect, great efforts were put into understanding the interplay between the chemical exchange, lifetimes of transient complexes, and nuclear spin interactions - much of the SABRE effect is determined by these factors[18–22]. As the spin interactions are not easy to change, many efforts focused on modifying the exchange and lifetimes. The mainstream way of doing this is to vary the composition of the mixture, solvent, temperature, and ligands surrounding the Ir center[23–29].

The SABRE complex lifetimes can be assessed indirectly by observing the effect of chemical exchange on polarization transfer efficiency[30,31]. Alternatively, a more direct measurement can be achieved using proton ($^1$H) exchange spectroscopy (EXSY)[32–35]. Following this, in a previous study[36], we have compared the performance of $^1$H 1D SEXSY[37–39] and 2D EXSY[32,34,35] NMR spectroscopic techniques and the robustness of the extracted exchange rate constants for the most widely used SABRE organometallic complex (IrIMes). However, these techniques exploit the thermal $^1$H NMR spectra with a lower spectral dispersion than heteronuclear NMR[40–42]. For example, measuring acetonitrile and metronidazole SABRE complexes' lifetimes with $^1$H NMR was impossible because of the marginal chemical shift differences between the corresponding free and catalyst-bound substrate species. Both compounds are common SABRE targets, and the latter one is an antibiotic and prospective hypoxic radiosensitizer whose MRI properties were recently explored[43]. $^{15}$N-enhanced EXSY also uses the long relaxation of $^{15}$N compared to $^1$H, better-separating exchange and relaxation time periods, allowing for a more precise measurement of relatively slow exchange.

Our study aims to quantify the lifetimes of SABRE-active Ir complexes whilst improving the precision of such estimates. Furthermore, we want to increase the scope of the systems where such analysis is possible, especially, for example, in congested spectra where the selective excitation of $^1$H resonances of interest is impossible; such situations are common at low magnetic fields. To achieve this aim, we modified a high-field $^{15}$N hyperpolarization scheme that has been used before for both Crabtree's catalyst[32] and N-heterocyclic carbene complex studies[40]. The detection of the insensitive $^{15}$N nucleus is facilitated by polarization transfer from the pH$_2$-derived hydride ligands via a phINEPT+ spin order transfer (SOT) sequence[44]. This process boosts the sensitivity of $^{15}$N EXSY and allows to rapidly acquire the spectra benefiting from the hyperpolarization and chemical shift dispersion of $^{15}$N.

When such INEPT-type sequences are used for pH$_2$-based heteronuclear NMR signal enhancement[32,40], up to 50% X-nuclei polarization has been predicted theoretically[45–47]. The efficiency of this approach, though, can be improved further if frequency-selective radiofrequency (RF) pulses are used instead of hard pulses to excite the protons originating from pH$_2$[14,46,48]. Such an approach is not applicable when these protons are chemically equivalent, which is the case for symmetric Ir complexes that only yield PHIP due to magnetic inequivalence associated with the AA′X or AA′XX′ type spin system created with $^{15}$N labeling[49]. More recently, the ESOTHERIC SOT pulse sequence has been introduced for hydrogenative PHIP[50]. This SOT sequence offered up to 100% X-nuclear polarization using only broadband RF pulses.

Here, we compare the efficacy of the ESOTHERIC and phINEPT+ approaches for $^{15}$N SABRE and use them to elucidate substrate exchange rates. Experimentally, the ESOTHERIC approach provided up to 35% higher polarization levels compared to phINEPT+. Using the larger signal enhancement of SABRE-ESOTHERIC, the lifetimes of the SABRE-active Ir complexes were studied with greater accuracy.

## Results and discussion
### Spin order transfer sequences
phINEPT+ and ESOTHERIC sequences (Fig. 2A, B) were used to transfer polarization from the pH$_2$-derived hydride ligands of the SABRE catalyst to the $^{15}$N nuclei of the bound substrate. Although here only symmetric spin systems were studied, resulting in two chemically equivalent hydrides IrHH, the fact that these nuclei are strongly magnetically nonequivalent results in an averaging of the singlet state, leaving imbalanced observable antiphase $\hat{I}_z\hat{S}_z$ spin order[51,52]. The discussed pulse sequences are designed to transfer this type of polarization to heteronuclei.

Adding two 90° $^{15}$N pulses after the SOT block allows for observing the chemical exchange of the substrate (Fig. 2C). Here, the first 90° pulse after the SOT block converts the transverse net $^{15}$N magnetization into longitudinal magnetization (as was suggested for $l$-phINEPT+)[40]. During the following free evolution interval $\tau_e$, some of the equatorial substrate ligands dissociate from the complex. The second 90° pulse samples the longitudinal magnetization of the free and bound substrate[53]. By variation of $\tau_e$, the exchange kinetics are monitored, which can be analyzed to find the lifetime of the SABRE complex. We refer to these sequences as SABRE-INEPT and SARBE-ESOTHERIC.

### SABRE with phINEPT+ and ESOTHERIC
The SOT efficiency in SABRE experiments is not trivial to predict due to a complex interplay of the chemical exchange and spin-spin interactions[12,31]. Therefore, first, we estimated the optimum parameters for phINEPT+ and ESOTHERIC for polarization transfer from pH$_2$-derived hydrides to the equatorially bound substrate (eS) on the SABRE catalyst. For this, we experimentally varied the two intervals $\tau_1$ and $\tau_2$ (Fig. 2A, B) and measured the $^{15}$N signal of the bound $^{15}$N-labeled substrates at 288 K for $^{15}$N-Py, $^{15}$N-NAM, $^{15}$N-4AP, and $^{15}$N-ACN and at 267 K for $^{15}$N$_3$-MNZ (Fig. 3). At 288 K, $^{15}$N$_3$-MNZ did not provide high polarization values, as will be evident below, due to the rapid dissociation rate—one of the fastest among the tested substrates. Therefore, we cooled the sample down to 267 K where the

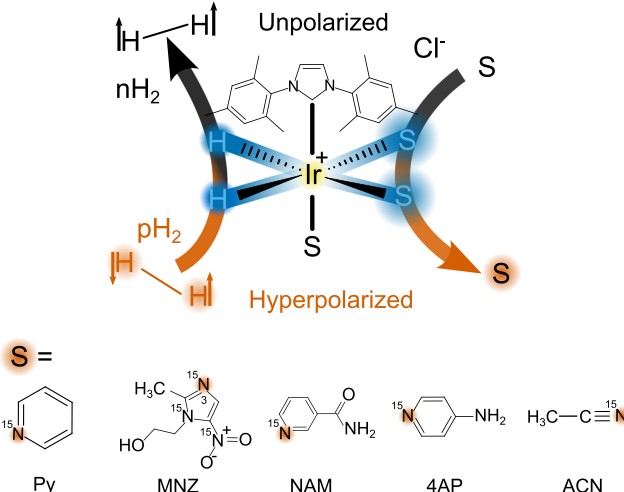

**Fig. 1 | Schematic diagram for SABRE hyperpolarization.** The pH$_2$ and substrate (S) molecules used here transiently bind to a SABRE complex of the form [Ir(H)$_2$(IMes)S$_3$]Cl. At the high magnetic field, spin-spin interactions together with RF pulses drive polarization transfer from parahydrogen (pH$_2$, vermillion) into the substrate, resulting in the hyperpolarization of both equatorial substrate ligands (blue) and, ultimately, the free substrate (vermillion) and formation of normal hydrogen (nH$_2$, black). (Py pyridine, MNZ metronidazole, 4AP 4-aminopyridine, NAM nicotinamide, ACN acetonitrile).

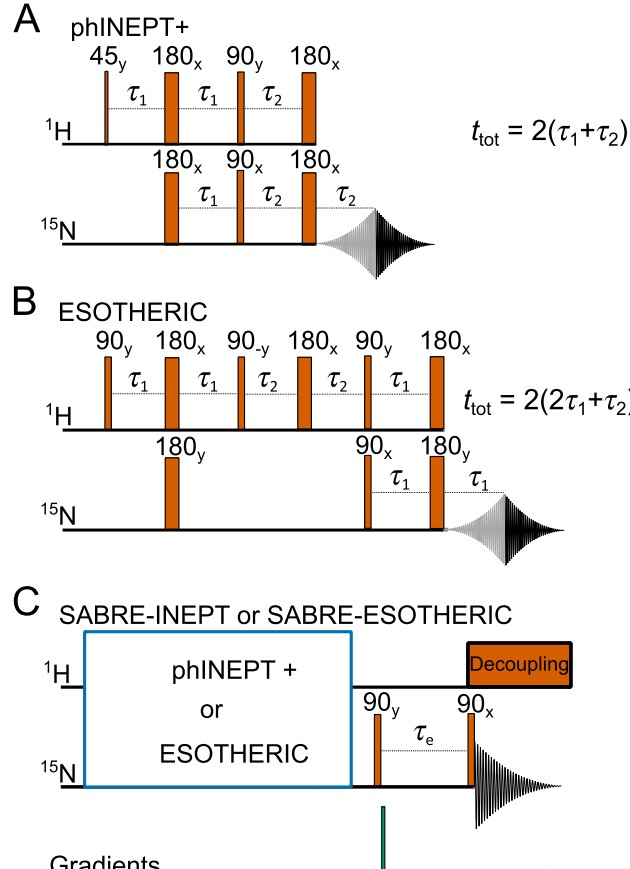

**Fig. 2 | SOTs for SABRE hyperpolarization and measurement of chemical exchange.** phINEPT+ (**A**) and ESOTHERIC (**B**) SOT sequences and the modifications allowing investigation of chemical exchange (**C**, SABRE-INEPT or SABRE-ESOTHERIC). The phINEPT+ (**A**) and ESOTHERIC (**B**) sequences convert the spin order of the $pH_2$-derived hydride ligands into transverse (X) magnetization on the $^{15}N$ nuclei of the bound substrate. Adding two 90° $^{15}N$ RF pulses enables the observation of chemical exchange between bound and free substrates during the mixing time $\tau_e$ between the pulses (**C**). The blue square in (**C**) should be substituted with an appropriate SOT sequence: here phINEPT+ or ESOTHERIC. The total phINEPT+ time $t_{tot} = 2(\tau_1 + \tau_2)$ and the total ESOTHERIC time $t_{tot} = 2(2\tau_1 + \tau_2)$.

dissociation rate sufficiently slowed down, enabling polarization of 0.8% at optimal SOT parameters. We used a 1 ms time grid to assess the optimal conditions that required from 25 to 81 experiments per map, with 453 hyperpolarization experiments in total. The optimal SOT parameters found through this approach were used in all other experiments, including different temperatures for $^{15}N_3$-MNZ and other substrates: this is justifiable because although exchange rates are changing significantly as a function of temperature, the spin-spin interactions—the driving force for SOT—do not change significantly in the studied range.

ESOTHERIC always provided from 3 to 35% higher polarization than phINEPT+ (Fig. 3). For example, a maximum $^{15}N$ polarization of 0.96% was achieved with phINEPT+ ($\tau_1 = 11$ ms, $\tau_2 = 9$ ms, $t_{tot} = 40$ ms) and 0.99% with ESOTHERIC ($\tau_1 = 7$ ms, $\tau_2 = 6$ ms, $t_{tot} = 40$ ms) for $^{15}N$-ePy (Fig. 3A). The largest relative signal gain of 35% was achieved for 4AP: phINEPT+ provided 0.14% ($\tau_1 = 14$ ms, $\tau_2 = 9$ ms, $t_{tot} = 46$ ms), while ESOTHERIC yielded 0.19% ($\tau_1 = 8$ ms, $\tau_2 = 3$ ms, $t_{tot} = 38$ ms) $^{15}N$ polarization. 1% of $^{15}N$ polarization corresponds to a signal enhancement factor of 2584 at 9.4 T. Although the ESOTHERIC sequence has more RF pulses and time intervals, the total duration of the phINEPT+ sequence is not shorter than that of ESOTHERIC, when both use the optimal settings for the associated

chemical system (Fig. 3): an additional positive feature of ESOTHERIC sequence.

The spectral lines of hyperpolarized molecules measured immediately after phINEPT+ and ESOTHERIC have distorted line shapes due to polarization of zero and higher-order quantum coherences, which are populated due to numerous spin–spin interactions in the system. These coherences can be suppressed by adding a corresponding quantum filter[54]. In the following experiments with SABRE-INEPT and SABRE-ESOTHERIC (SOT scheme in Fig. 2C), we used a pulsed field gradient to filter out quantum coherences of higher order, which was found to be sufficient to provide distortionless lines (Fig. S1, SI).

As mentioned, ESOTHERIC is expected to give twice the signal intensity of phINEPT+, however, only at specific conditions and for a three-spin-½ system. Here, the spin systems are more complex (>4 spins), and additional interactions with other magnetic nuclei (e.g., $^1H$ or $^{15}N$ in our case) in the substrates destructively modulate SOT. The application of selective refocusing pulses can improve the efficiency of both sequences since they reduce the system to an effective four-spin system (two hydrides IrHH and two equatorial $^{15}N$ nuclei); however, it was not tested here.

Note that experiments for all compounds were conducted at ambient pressure except for experiments with MNZ. This is because preliminary tests at ambient pressure with MNZ revealed the necessity for a long activation period, exceeding one hour and did not provide sufficient signal gain for reliable measurements. Using a higher pressure (6.6 bar), a common practice to increase activation rate and polarization yield[55], the sample was activated within 40 minutes, and both phINEPT+ and ESOTHERIC provided close to 0.8% $^{15}N$ polarization.

After activation, we conducted experiments with the same sample for up to 5 h. During the course of the experiment, the sample volume decreased by less than 10% thanks to the low temperatures used. Still, this leads to additional errors when evaluating the ligand exchange rate constants. To avoid this, two solutions were proposed in literature (not used here): adding a sample reservoir[56] that keeps the sample volume constant or saturation of dry $pH_2$ with methanol that significantly slows down evaporation[29,57]. Note that the latter approach works thanks to the slow para-to-ortho conversion of $H_2$ in clean solvents and tubes[58].

## Transferring polarization to the free substrate

Both ESOTHERIC and phINEPT+ SOT sequences create transverse net magnetization on the equatorial substrate ligands. Chemical exchange can therefore transfer a bound response to signal for free substrate. However, to measure such a process beyond the decay time of transverse magnetization ($T_2^*$) it is necessary to add a 90° RF pulse that converts the transverse magnetization of the bound substrate to its longitudinal magnetization. The subsequent chemical exchange transfers it to the free substrate that can be measured by applying another 90° pulse after the appropriate free evolution interval $\tau_e$ between the two pulses (Fig. 2C).

The chemical exchange was monitored using SABRE-ESOTHERIC (or SABRE-INEPT) for the same five compounds at several temperatures, using optimized SOT parameters (intervals $\tau_1$ and $\tau_2$). At $\tau_e = 0$ s, the polarization of the free substrate is zero, and that of the bound form is at its maximum. Increasing the time delay between the last two pulses encodes polarization transfer from the bound to the free substrate. For example, the signal of initially hyperpolarized equatorial $^{15}N$-NAM ($^{15}N$-eNAM) with a chemical shift of 255.7 ppm was decreasing while that of free $^{15}N$-NAM ($^{15}N$-fNAM) at 302.3 ppm was initially increasing as a function of $\tau_e$ (Fig. 4). After about 1 s, visible polarization reaches its maximum on the free form and then decreases due to relaxation.

## Exchange rates obtained with SABRE-ESOTHERIC

All kinetics for longitudinal magnetization of free ($M_f$) and equatorial ($M_e$) substrates measured with SABRE-ESOTHERIC (one example for NAM is given in Fig. 4) were fitted using global fitting for exponential decay

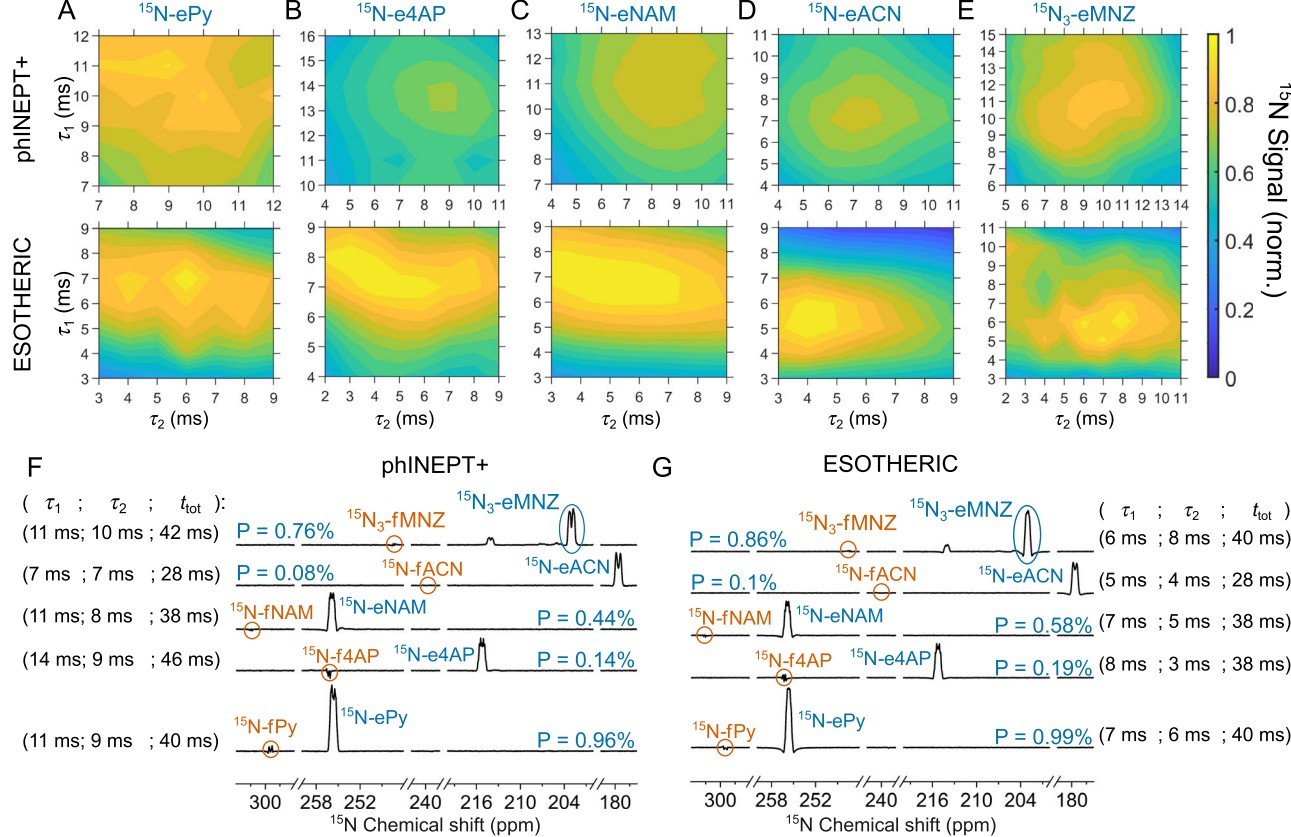

**Fig. 3 | Performance of phINEPT+ and ESOTHERIC SOT sequences.** Experimentally measured $^{15}$N polarization of [$^{15}$N]pyridine ($^{15}$N-Py, **A**), 4-amino[$^{15}$N] pyridine ($^{15}$N-4AP, **B**), [$^{15}$N]nicotinamide ($^{15}$N-NAM, **C**), [$^{15}$N]acetonitrile ($^{15}$N-ACN, **D**), and [$^{15}$N$_3$]metronidazole ($^{15}$N$_3$-MNZ, **E**; the signal of $^{15}$N-3 site of the symmetric [Ir(IMes)(H)$_2$(MNZ)$_3$] complex with resonance at 203 ppm was analyzed) as a function of $\tau_1$ and $\tau_2$ delays of phINEPT+ (**A**–**E** top) and ESOTHERIC (**A**–**E** bottom) SOT sequences at 9.4 T. The measurements were performed at 267 K

for $^{15}$N$_3$-MNZ and 288 K for the other substrates. phINEPT+ and ESOTHERIC 2D maps were jointly normalized for each substrate (i.e., the signals should not be compared between the different substrates). $^{15}$N NMR spectra with the maximum polarization of the substrate at the equatorial site and the corresponding $\tau_1$ and $\tau_2$ values are given for phINEPT+ (**F**) and ESOTHERIC (**G**). At optimum conditions, ESOTHERIC provided higher polarization than phINEPT+. e and f stand for equatorial and free substrates.

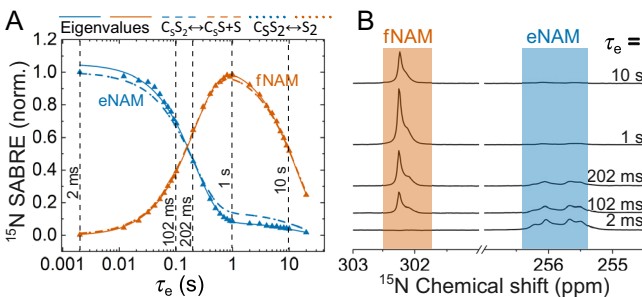

**Fig. 4 | Dissociation kinetics of equatorially bound NAM.** Normalized $^{15}$N signals (**A**) and $^{15}$N NMR spectra (**B**) of free ($^{15}$N-fNAM, vermillion) and equatorial ($^{15}$N-eNAM, blue) $^{15}$N-NAM as a function of free evolution interval $\tau_e$ at 293 K and 9.4 T. $^{15}$N signals were enhanced with ESOTHERIC with $\tau_1 = 7$ ms and $\tau_2 = 5$ ms. Kinetics in **A** were fitted with a shared biexponential decay function (eigenvalues analysis, resulting in two decay rates of $k = (4.63 \pm 0.07)$ s$^{-1}$ and $R = (0.07 \pm 0.002)$ s$^{-1}$), using models $C_SS_2 \leftrightarrow C_SS + S$ and $C_SS_2 \leftrightarrow S_2$[36]. The slower rate constant $R$ corresponds to the effective relaxation rate, while the faster rate constant $k$ corresponds to the effective exchange rate.

constants $R$ and $k$ in Eq. 1 (see Ref. 36 for details).

$$M_{e,f} = A_{e,f}e^{-Rt} + B_{e,f}^{-kt} + C_{e,f} \qquad (1)$$

where e, f stand for equatorial and free substrates, giving eigenvalues of two-site exchange: $R = 0.072 \pm 0.002$ s$^{-1}$ and $k = 4.63 \pm 0.07$ s$^{-1}$ for exchange of NAM at 293 K (Fig. 4A). As a result, we have two decay constants: the smaller one (almost temperature independent) attributed to the effective relaxation rate constant $R$ and the larger one (strongly temperature dependent) attributed to the exchange rate constant $k$ which lets us calculate the dissociation rate $k_d$ using Eq. 2 [36].

$$k_d = \frac{k}{0.5 + \frac{[C_SS_2]}{[S]}} \qquad (2)$$

The concentration ratio $\frac{[C_SS_2]}{[S]}$ was calculated from the $^1$H spectrum. For $\frac{[C_SS_2]}{[S]} = 12.24$, this yields $k_d = 8.0 \pm 0.1$ s$^{-1}$.

Additionally, the resulting data were fit to two chemical exchange models developed in the earlier work[36]: the more complete model $C_SS_2 \leftrightarrow C_SS + S$ and the simplified model $C_SS_2 \leftrightarrow S_2$. In this example (Fig. 4A), both models yielded the same $k_d$ within the error margin $k_d = 10.0 \pm 0.3$ s$^{-1}$.

Dissociation rates for all substrates were calculated using these three models (Table S3, SI) and plotted as functions of inverse temperature together with the weighted mean value across the models (Fig. 5). The mean dissociation rate constants were further fitted using Eyring equation (Eq. S8, SI).

The dissociation rates for the five substrates under study were significantly different. $^{15}$N-ACN and $^{15}$N$_3$-MNZ exchanged significantly faster

than the pyridine derivatives ¹⁵N-Py, ¹⁵N-4AP, and ¹⁵N-NAM. It has been predicted that, for pyridine derivatives, in the absence of steric effects, the lower the pKa of the conjugate acid, the stronger the ligand binding[59]. ACN and MNZ do not belong to the six-membered heterocycle family; the higher charge density of six-membered heterocycles leads to stronger interactions between charged entities (Ir⁺ and the ring) and slows down dissociation rates. In addition, MNZ has additional steric hindrances. A rationale for the rapid loss of ¹⁵N-ACN (the smallest molecule in the group) needs further evaluation, although these data are consistent with other studies, as shown in Ref. 60. To confirm the influence of the reagent structure, a larger pool of substrates, including five-membered rings with 1–3 nitrogens, should be tested.

With SABRE-ESOTHERIC, the lifetimes obtained from the $C_SS_2 \leftrightarrow C_SS + S$ and $C_SS_2 \leftrightarrow S_2$ models were almost equal but differed from those derived from the eigenvalues analysis. Comparison with the previously published ¹H SEXSY data[36] for Py, 4AP, and NAM showed mixed results. For Py, SABRE-ESOTHERIC typically provided greater dissociation rate constants than ¹H SEXSY with up to a 2.4-fold difference (Table S3). For 4AP and NAM, the values obtained using these two approaches were closer.

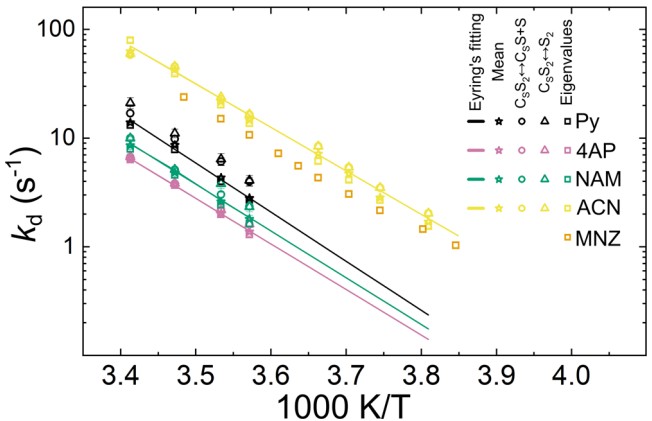

**Fig. 5 | Substrate dissociation rate constants extracted from SABRE-ESOTHERIC kinetics as a function of temperature.** Dissociation rate constants for ¹⁵N-Py (black), ¹⁵N-4AP (reddish purple), ¹⁵N-NAM (bluish green), ¹⁵N-ACN (yellow), and ¹⁵N₃-MNZ (orange) were obtained using model $C_SS_2 \leftrightarrow C_SS + S$ (circles), model $C_SS_2 \leftrightarrow S_2$ (triangles), and using eigenvalues analysis (squares). The weighted mean $k_d$ values (stars) were calculated using Eq. S6 (SI). The mean $k_d$ values were fitted using Eyring equation (line).

Due to the negligible ¹H chemical shift variation of MNZ and ACN upon coordination with the Ir catalyst, we could not measure their exchange rates with ¹H EXSY before. Here, however, we were able to determine these values because of the larger ¹⁵N chemical shift difference of 45–60 ppm between equatorial-bound and free substrates.

In the case of ¹⁵N₃-MNZ, several Ir complexes are present in the SABRE mixture, as confirmed by ¹⁵N and ¹H hyperpolarized NMR (Figs. S8 and S9, SI). The major complexes are [Ir(IMes)(H)₂(MNZ)₃], [Ir(IMes)(H)₂(MNZ)₂Cl], and [Ir(IMes)(H)₂(MNZ)₂(CD₃OD)] species[29]. The precise assessment of the substrate exchange, considering all these complexes, requires a more detailed analysis of the exchange mechanism and, thus, will be a subject of our future studies. Here, we simplified the matter by estimating phenomenological exchange rates between all hyperpolarized equatorial-bound ¹⁵N₃-MNZ species and free ¹⁵N₃-MNZ. Note, however, that the $\tau_1$ and $\tau_2$ delays in SABRE-ESOTHERIC SOT sequence were optimized for the polarization transfer in [Ir(IMes)(H)₂(MNZ)₃] species (see Fig. S10). Therefore, this species has the primary contribution to the overall observed substrate dissociation flux. The fitting of the total dissociation flux of MNZ with models $C_SS_2 \leftrightarrow C_SS + S$ and $C_SS_2 \leftrightarrow S_2$ was not accurate, which is reasonable considering these simplifications. However, the curves fit well with a simple two-exponential decay (Fig. S6, SI). Therefore, only this approach was used to estimate the MNZ dissociation rate constants, enthalpy and entropy of activation.

## Enthalpy ΔH‡ and entropy ΔS‡ of activation

Activation enthalpies $\Delta H^\ddagger$ and entropies $\Delta S^\ddagger$ were calculated using the dissociation rate constants $k_d$ (Fig. 6, Table S5, SI). The better signal-to-noise ratio resulting from the SABRE-ESOTHERIC approach enabled greater accuracy in the ensuing rate constants and, therefore, more accurate thermodynamic data to be obtained (Table S5, SI). The $\Delta H^\ddagger$ values obtained were ~70–80 kJ/mol for all substrates under study (Table S5, SI). This value reflects the bond strength as the process of ligand loss is dissociative. The smaller substrates Py and ACN showed $\Delta S^\ddagger$ of ca. 60 J/(mol K) according to eigenvalues analysis (Table S5). 4AP and NAM had $\Delta S^\ddagger$ of ca. 40 J/(mol K), and MNZ had $\Delta S^\ddagger = 29$ J/(mol K) (although it should be noted that variation of $\Delta S^\ddagger$ values between the $C_SS_2 \leftrightarrow C_SS + S$ and $C_SS_2 \leftrightarrow S_2$ models, eigenvalues analysis and mean $k_d$ values was quite high).

To assess the measured enthalpies of activation, we computed dissociation pathways and their energies using density functional theory (DFT) quantum chemistry simulations at the B3LYP-D4/def2-TZVP level of theory[61–64] including DLPNO-CCSD(T)/def2-TZVP single point energies with implicit solvation. The ligand dissociation potential energy surface did

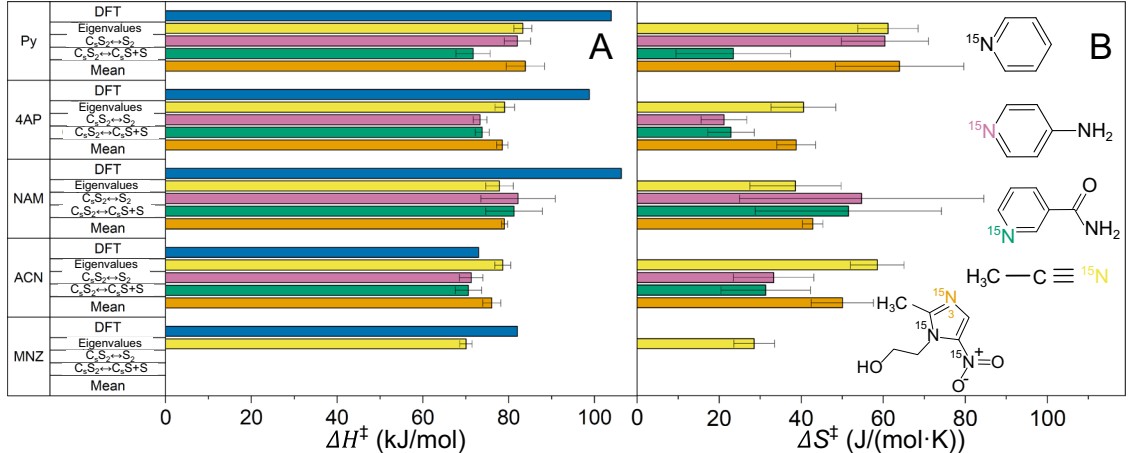

**Fig. 6 | Comparison of estimated enthalpy and entropy values.** Enthalpies $\Delta H^\ddagger$ (**A**) and entropies $\Delta S^\ddagger$ (**B**) of activation for the corresponding IrIMes complexes with ¹⁵N-Py, ¹⁵N-4AP, ¹⁵N-NAM, ¹⁵N-ACN, and ¹⁵N₃-MNZ as a substrate. $\Delta H^\ddagger$ and $\Delta S^\ddagger$ were obtained from the dissociation rate constants obtained with the model $C_SS_2 \leftrightarrow C_S S + S$ (bluish green), with the model $C_SS_2 \leftrightarrow S_2$ (reddish purple), eigenvalues analysis (yellow), mean $k_d$ values (orange), and from DFT calculations (blue). The values (except those obtained from DFT calculations) are obtained by fitting $k_d$ (Table S3, Fig. S7, SI) to the Eyring equation and are given in Table S5.

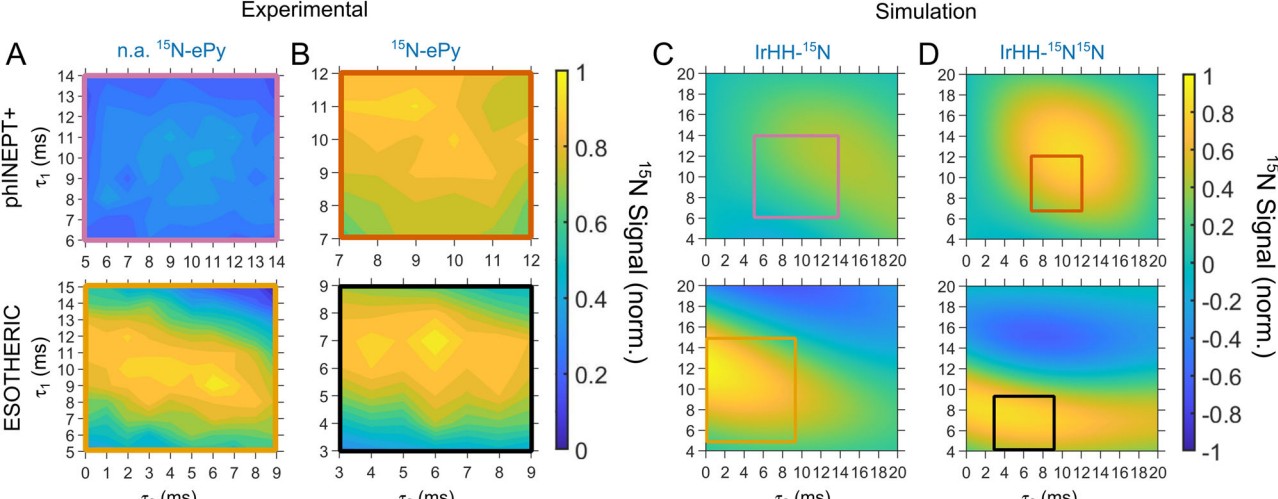

**Fig. 7 | Comparison of phINEPT+ and ESOTHERIC SOT for $^{15}$N-labeled and unlabeled substrate.** Performance of phINEPT+ and ESOTHERIC SOT sequences for pyridine (n.a.—**A**, $^{15}$N—**B**) and simulations for three- (**C**) and four- (**D**) spin-½ systems. phINEPT+ and ESOTHERIC 2D maps were jointly normalized for each panel. Rectangles on **C**, **D** highlight the corresponding experimentally measured sections. phINEPT+ and ESOTHERIC SOT simulations have a maximum of 27.7% and 66.9% polarization for three-spin-½ systems (**C**) and 21.2% and 21.8% polarization for four-spin-½ systems (**D**): ESOTHERIC is 3 times more efficient for

three-spin-½ system than for four-spin-½ system. Parameters used for the simulations were: $J_{HH} = -8$ Hz, $J_{NHt} = -22$ Hz, and $J_{NHc} = 0$ Hz. To consider exchange during SOT, all polarization values were multiplied by the $\exp(-k_d \times t_{tot})$ value, with $k_d = 2.7$ s$^{-1}$ and $t_{tot}$ is the total time of the SOT sequence, which accounts for the fact that only a fraction of molecules that survived whole SOT sequence step contribute to the final NMR signal; alternatively, one can use an explicit simulation of chemical exchange and spin evolution[20,21].

not exhibit an identified transition state (Fig. S12, SI). Hence, we used the computed dissociation enthalpies to estimate $\Delta H^{\ddagger}$ (Fig. 6, Table S6, SI). A qualitative agreement is observed between the computed dissociation energies and the enthalpies of activation derived from the measured rate constants.

ACN exhibited the smallest dissociation energy of 73 kJ/mol, MNZ had a higher dissociation energy of 82 kJ/mol, and Py, 4AP, and NAM fall into a group of similar values of 104, 99, and 106 kJ/mol (Table S6). Thus, the actual enthalpies seem to be overestimated in the computational values, very likely due to approximations related to the implicit description of solvation (for computational values see Supplementary Discussion: Quantum chemical calculations).

In recent work[65], rate constants $k_d$ of Py and NAM dissociation were found to be $k_d = 10.8$ s$^{-1}$ and $k_d = 7.86$ s$^{-1}$ at 298 K, respectively. These values were determined using the $^1$H EXSY NMR exchange spectroscopy technique. Corresponding transition-free Gibbs energy at 298 K for losing Py and NAM was estimated to be approximately 67.05 and 68.22 kJ/mol using the Eyring plots of the dissociation rates[65]. DFT estimates are 45.48 kJ/mol for Py and 68.22 kJ/mol for NAM[65]. Here, DFT failed to replicate the similarities between the two substrates measured experimentally.

In our study, we estimated the exchange rate constants of Py and NAM at 298 K to be 24.32 and 15.68 s$^{-1}$, respectively, with experimentally estimated activation enthalpies of $84 \pm 4$ and $79 \pm 1$ kJ/mol and DFT calculated values of 104 and 106 kJ/mol, respectively. Our energies are somewhat higher than those reported before[65], however, the level of theory applied herein successfully reproduces similarities in activation energies with only a slight offset.

The difference between exchange rate constants measured here and in Ref. 65 is almost two-fold, likely as a result of the different definitions of the exchange rate. Based on our previous work[36], we define the $k_d$ value in such a way that the lifetime of the Ir complex is given by $1/k_d$. Because the exchange proceeds via a dissociative or $S_N1$-type mechanism and two equatorial substrates dissociate at the same rate, the individual dissociation rate of a chosen single ligand is half the dissociation rate of any of the two ligands. Therefore, when comparing the results presented here with the literature values, special attention should be paid to the description of the exchange models. After correcting for this factor, the dissociation rate constant values are similar.

## Hyperpolarization of naturally abundant (n.a.) pyridine

To assess the hyperpolarization efficiency, we attempted to hyperpolarize unlabelled 400 mM pyridine (the natural abundance of $^{15}$N isotope is about 0.4%) with a 4 mM IrIMes catalyst. We experimentally transferred polarization to equatorial pyridine using both phINEPT+ and ESOTHERIC pulse sequences (Fig. 7A) and measured exchange rates (Fig. S11, SI) that gave deviation of up to 20% compared to values measured with labeled $^{15}$N-Py (Table S3). The maximum achievable polarization for equatorially bound pyridine was ~7.2% using phINEPT+ and ~17.5% using ESOTHERIC: ESOTHERIC provided twice the signal than phINEPT+. This is because the system was reduced effectively from a four-spin system consisting of two IrHH protons and two $^{15}$N nuclei of two equatorial substrates to the system with only one $^{15}$N nucleus in the equatorial plane. In this case, ESOTHERIC performance is optimal, while phINEPT+, at best, can give 50% of ESOTHERIC polarization values. To confirm our measurements, we simulated SOT performance for three- and four-spin systems without chemical exchange to estimate performance in the case of n.a. Py (Fig. 7C) and $^{15}$N-Py (Fig. 7D). The data qualitatively agree with experimental observation.

## Conclusions

The approach based on polarization transfer to heteronuclei and then the encoding of an exchange rate illustrated here for $^{15}$N reflects an exciting addition to the toolkit of mechanistic chemists. We compared the performance of broadband phINEPT+ and ESOTHERIC SOT sequences for transferring nuclear spin polarization from the parahydrogen-derived hydride ligands of the Ir SABRE catalyst to the $^{15}$N nuclei of the coordinated substrates examplified by $^{15}$N-labeled pyridine, 4-aminopyridine, nicotinamide, acetonitrile, metronidazole and none labeledpyridine. ESOTHERIC was shown to be superior, providing higher $^{15}$N polarization values, especially in the none labeled case. The created hyperpolarization allowed the subsequent measurement of substrate exchange rates by $^{15}$N NMR. The chemical exchange models $C_SS_2 \leftrightarrow C_SS + S$ and $C_SS_2 \leftrightarrow S_2$ described in the literature[36] were used to achieve this. More precise data fitting provided by strong $^{15}$N NMR signal enhancement allowed us to estimate the dissociation rate constants more robustly and to measure enthalpies of activation with higher accuracy than in the earlier work. The utility of $^{15}$N detection offers much better chemical shift dispersion to

delineate the free and catalyst-bound species in SABRE mixtures compared to conventional 1D $^1$H SEXSY or 2D $^1$H EXSY techniques. As a result of the greatly enhanced spectral resolution, it became possible to measure the dissociation rate constants for acetonitrile and metronidazole. The reported methodology broadly applies to various substrates suitable for SABRE hyperpolarization.

Through this methodology, it would be possible to evaluate ligand dissociation kinetics in the wide-ranging number of transition metal[66] and metal-free[67] species detected through PHIP. The suggested approach is not limited to $^{15}$N and can be used for other magnetic nuclei. For example, one can study dissociation of phosphine ligands (which are common in organometallic catalysis) using $^{31}$P NMR. Furthermore, as $^{31}$P is essentially 100% abundant, labeling is not needed, and it may be possible to extend these approaches to normal NMR without hyperpolarizsation using regular INEPT. Exchange of carbon-containing ligands may be studied as well, in this context, the 1% natural abundance suggests hyperpolarization will be necessary. For example, recently, we measured *J*-coupling constants between IrHH protons and coordinated [1-$^{13}$C]pyruvate using a modified INEPT sequence[14]. A slight modification of the sequence presented here will allow one to gain insight into the chemical kinetics of pyruvate exchange during SABRE. This approach is also not limited to RF SOT-driven hyperpolarization. With demonstrated $^{15}$N polarization values achieved in ultra-low field experiments[68], kinetics measurements even on low-field benchtop NMR systems should be possible. As the accuracy of this approach has been shown to be high, the results will be significant.

## Methods
### Chemicals
Perdeuterated Ir precatalyst [Ir-$d_{22}$] = [IrCl(COD)(IMes-$d_{22}$)] was synthesized according to Ref. 55 (IMes = 1,3-bis(2,4,6-trimethylphenyl)imidazol-2-ylidene, COD = 1,5-cyclooctadiene). [$^{15}$N]Pyridine ($^{15}$N-Py, 486183, Sigma-Aldrich), [$^{15}$N]acetonitrile ($^{15}$N-ACN, 487864, Sigma-Aldrich), and methanol-$d_4$ (441384, Sigma-Aldrich) were used as received. 4-Amino[$^{15}$N] pyridine ($^{15}$N-4AP) was synthesized according to Ref. 69. [$^{15}$N]Nicotinamide ($^{15}$N-NAM) was synthesized in a two-step reaction from unlabeled NAM (72340, CAS: 98-92-0, Sigma-Aldrich) via the corresponding Zincke salt followed by nitrogen exchange with $^{15}$NH$_4$Cl (299251, CAS: 39466-62-1, Sigma-Aldrich)[70,71]. [$^{15}$N$_3$]metronidazole ($^{15}$N$_3$-MNZ) was synthesized according to Ref. 72.

### Sample preparation
The samples were prepared by mixing one of the five SABRE substrates ($^{15}$N-Py, $^{15}$N-NAM, $^{15}$N-4AP, $^{15}$N$_3$-MNZ, or $^{15}$N-ACN) at a concentration of 40, 80, 80, 80, and 120 mM, respectively, with 4 mM of [Ir-$d_{22}$] in 1.2 mL of methanol-$d_4$.

### Parahydrogen enrichment
pH$_2$-enriched hydrogen gas with a 93% fraction of pH$_2$ was prepared by passing high-purity hydrogen over hydrated iron(III) oxide at 25 K using a pH$_2$ generator similar to the one in Ref. 73. For 52% content, pH$_2$ was prepared using an in-house built liquid nitrogen system described before in Ref. 74.

### phINEPT+, ESOTHERIC, SABRE-INEPT, and SABRE-ESOTHERIC experiments
All experiments with hyperpolarization were carried out on a Bruker 400 MHz spectrometer using a 5 mm BBO probe. Typically, a 600 μL sample was transferred to a screw-cap 5 mm NMR tube (507-TR-8, Wilmad Lab). The NMR tube was then attached to a low-pressure system, which enabled pH$_2$ bubbling through the solution using 0.2-0.3 bar overpressure with respect to the ambient pressure[75], and placed inside the spectrometer. First, we flushed the sample with pH$_2$ for 30 min to activate the catalyst. In the case of $^{15}$N$_3$-MNZ, 400 μL of the sample was instead transferred to a high-pressure 5 mm NMR tube (524-PV-7, Wilmad Lab) connected to a high-pressure system operating at 6.6 bar (similar to the one used in Ref. 76).

The $^{15}$N$_3$-MNZ sample was bubbled with pH$_2$ for 40 min to activate the catalyst precursor. Then, a series of experiments with one of the SOT sequences: phINEPT+, ESOTHERIC, SABRE-INEPT, or SABRE-ESO-THERIC, were carried out. One experiment consisted of (1) 10 s period of relaxation delay followed by (2) 2 s of WALTZ-16 on $^1$H channel, then (3) 10 s of pH$_2$ bubbling through the sample, followed by (4) a delay of 1.5 s for bubbles to disappear, followed by (5) one of the SOT sequences (Fig. 2). The following parameters of RF pulses were used: for $^1$H, the frequency of the RF pulse was −22.79 ppm in resonance with hydride protons, and the duration of hard 90° RF pulse was 11.575 μs; for, $^{15}$N the frequency was in resonance with the corresponding equatorial substrate and the duration of hard 90° RF pulse was 21.25 μs.

### Data processing
All spectra were analyzed using spectral data analyzing software Bruker TopSpin (4.1.4), MestReNova (14.2.2), and Origin (2021). SABRE-INEPT and SABRE-ESOTHERIC data were modeled using Origin or MATLAB (R2021a) as described in the text as well as in the previous publication[36].

### Data availability
Additional experimental and data analysis details, exchange rate constants and corresponding biexponential fittings of kinetic data, enthalpies and entropies of activation, analysis of SABRE complexes with MNZ (Supporting information), quantum chemical calculations (PDF), raw data, simulation Matlab scripts, and DFT calculated geometries of complexes and dissociation geometries can be accessed via Zenodo.

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

## Acknowledgements

A.N.P., C.A., and J.B.H. acknowledge funding from German Federal Ministry of Education and Research (BMBF) within the framework of the e: Med research and funding concept (01ZX1915C), DFG (PR 1868/3-1, PR 1868/5-1, HO-4602/2-2, HO-4602/3, GRK2154-2019, EXC2167, FOR5042, TRR287). MOIN CC was founded by a grant from the European Regional Development Fund (ERDF) and the Zukunftsprogramm Wirtschaft of Schleswig-Holstein (Project no. 122-09-053). O.G.S., N.V.C., and I.V.S. thank the Russian Science Foundation (grant 24-73-10093) for supporting the synthesis of $^{15}$N-4AP and $^{15}$N$_3$-MNZ, the development of exchange models, and data analysis. E.Y.C. thanks the funding support by NSF CHE-2404388. K.B. acknowledges funding from the DFG (BU 2694/6-1, BU 2694/9-1). A.N.P., C.A., K.B., M.P. and R.K. acknowledge support of DFG (469366436). S.B.D. is grateful to the UK Research and Innovation (UKRI), under the UK government's Horizon Europe funding guarantee [grant number EP/X023672/1], for funding. X.G. and A.A.A. would like to acknowledge funding by the Max–Planck–Gesellschaft and the MPI für Kohlenforschung.

## Author contributions

A.N.P.: conceptualization, C.A.: experiments, spin dynamics simulations, and investigation, X.G. and A.A.A. all quantum chemistry calculations, S.D., A.N.P., C.A., O.G.S., and J.B.H.: analysis, writing—original draft, discussion, A.N.P., and J.B.H.: supervision, funding acquisition, A.B. and R.H.: synthesis of $^{15}$N-NAM; N.V.C.: synthesis of $^{15}$N-4AP and $^{15}$N$_3$-MNZ. R.K., M.P., K.B., I.K., I.S., and E.C. have contributed to the preparation of the paper and all authors approved the final version of the paper.

## Funding

## Competing interests

E.Y.C. declares a stake of ownership in XeUS Technologies LTD and Vizma Life Sciences. E.Y.C. serves on the Scientific Advisory Board (SAB) of Vizma Life Sciences. The remaining authors declare no competing interests.
