## [Peer Review file · Communications Chemistry]

Analysis of Chemical Exchange in Iridium N-Heterocyclic Carbene Complexes Using Heteronuclear Parahydrogen-Enhanced NMR

Corresponding Author: Dr Andrey Pravdivtsev

Version 0:

Reviewer comments:

Reviewer #1

(Remarks to the Author)

Andrey N Pravdivtsev and co-workers introduced an experimental spin order transfer sequence where readout occurs at ^{15}N nuclei directly interacting with the SABRE catalyst. This topic is important as a lot of efforts at the field of SABRE hyperpolarization studies have focused on optimizing the polarization transfer step from the parahydrogen-derived hydride ligands to the substrate in SABRE.

In my view, this paper is worthy of publication as it is centred on the important facet of understanding SABRE and increasing hyperpolarization for a future improved method use. However, whilst reading through the manuscript, I found some aspects that were not very clear for me.

1. At the Introduction authors mentioned the newly developed method allowed to measure the SABRE exchange parameters for other substrates including $[1-^{13}\text{C}]\text{pyruvate}$. It should be provided citation if so. Also, the manuscript is describing study of ^{15}N -substrates and nothing regarding ^{13}C -substrates or especially pyruvate, perhaps it would be better not to mention it here.

2. At the Method section authors are describing samples preparation, but for me is not clear:

- Why for $\text{N}_3\text{-MNZ}$ has been used different conditions (high-pressure tube instead of low-pressure as for others, different time of catalyst activation)?
- Ref.48 should be checked as it doesn't refer to the text.
- What pressure has been used for other samples?
- Did pressure of parahydrogen bubbling make any difference for polarization level for different samples?
- Did the authors measure the sample volume after catalyst activation and after bubbling during experiments as well? It might be that after 30 - 40 min of bubbling, the methanol evaporates to give an optimised concentration which then becomes too concentrated and is affected detrimental to the polarization observed.

3. At the Results and Discussion section authors again talking about "experimentally varied the two-time intervals τ_1 and τ_2 and measured the ^{15}N signal of the bound ^{15}N -labeled substrates" at different temperatures for $\text{N}_3\text{-MNZ}$ and other samples.

- If this aspect concerning $\text{N}_3\text{-MNZ}$ has already been presented somewhere, it should be referred to, if not, then an explanation will need to be made in the current manuscript.
- The dissociation rates for the five substrates under study were significantly different. " $^{15}\text{N-ACN}$ and $^{15}\text{N}_3\text{-MNZ}$ exchanged significantly faster than the pyridine derivatives $^{15}\text{N-Py}$, $^{15}\text{N-4AP}$, and $^{15}\text{N-NAM}$ ". Do you have any explanation for it? Would be useful to provide them here.

4. In the Conclusion authors are saying,

- "Moreover, we envision that even at a natural abundance of ^{15}N nuclei with high enough polarization levels exceeding 50%, ^{66}Ir -complexes' lifetime can be studied using the chemical models presented here, even at low field benchtop NMR spectrometers." Perhaps it might make sense to test at least natural pyridine to confirm this, and maybe also test the current system on benchtop NMR.

Overall, the quality of the data, the attention to detail, make me believe the manuscript is ready for publication with some minor corrections.

Reviewer #2

(Remarks to the Author)

The manuscript highlights the advantage of using ^{15}N spin for analyzing chemical exchange kinetics in Iridium N-Heterocyclic Carbene complexes through heteronuclear parahydrogen-enhanced NMR techniques. Signal Amplification by Reversible Exchange (SABRE) is a nuclear spin hyperpolarization method that amplifies NMR signals and offers advantages over Dynamic Nuclear Polarization (DNP), another commonly used hyperpolarization technique. SABRE provides higher throughput and operates at a lower cost.

The manuscript introduces a novel ^{15}N NMR-based method, which enhances sensitivity in ligand exchange analysis, allowing for more accurate determination of exchange rates and thermodynamic parameters, including activation enthalpies and entropies. ^{15}N detection is especially advantageous for substrates where chemical shifts dispersion in ^1H NMR is minimal, such as in acetonitrile and metronidazole. The ^{15}N nucleus exhibits greater chemical shift dispersion, facilitating more precise quantification of exchange rates. Additionally, the manuscript compares two polarization transfer techniques for transferring longitudinal two-spin order from ^1H to ^{15}N . In comparing phINEPT+ and ESOTHERIC, the latter delivers up to 35% higher polarization levels, resulting in improved accuracy for measuring exchange kinetics and thermodynamic parameters of the complexes.

The phINEPT+ sequence is similar to the INEPT sequence, except for the inclusion of a 45-degree pulse that converts two-spin ^1H order into single-spin ^1H coherence. The subsequent transfer sequence utilizes a series of RF pulses, leveraging the scalar coupling between ^1H and ^{15}N spins. ESOTHERIC, on the other hand, converts longitudinal two-spin order into two-spin coherence (the sum of Zero and Double quantum coherence), utilizing two INEPT blocks to ultimately transfer coherence to the ^{15}N spin. While ESOTHERIC itself is not a new scheme, the application of Ramsey pulses to measure exchange rates represents an innovative approach in this work.

Overall, this manuscript elegantly explores spin physics with broad implications in biochemistry and medical imaging. I recommend the publication of the manuscript following revisions to the following points:

1. The description of $IzSz$ as purely singlet order may not hold if the two spins exhibit similar frequencies. The authors should clarify this point, as the protons in the study appear to have similar chemical shifts.
2. The ESOTHERIC sequence involves a longer pulse train than phINEPT. Does this affect the measurement of fast exchange rates?
3. There seems to be more phase distortion in the ESOTHERIC case compared to phINEPT, as shown in Figure 3. Can the authors clarify the origin of this distortion?
4. The phase cycling schemes used in ESOTHERIC and phINEPT should be described in greater detail.

Version 1:

Reviewer comments:

Reviewer #1

(Remarks to the Author)

I'm satisfied with updated version of the paper and think that provided results are really suitable to be published at the "Communications Chemistry".

Reviewer #2

(Remarks to the Author)

I was referring to the theory described in ChemistryOpen (2018, 7, 344–348), which uses longitudinal order as singlet order.

I am satisfied with the revisions made by the authors and recommend the manuscript for publication in its current form.

Reviewer #1:

Recommendation:

This topic is important as a lot of efforts at the field of SABRE hyperpolarization studies have focused on optimizing the polarization transfer step from the parahydrogen-derived hydride ligands to the substrate in SABRE.

Comments:

In my view, this paper is worthy of publication as it is centered on the important facet of understanding SABRE and increasing hyperpolarization for a future improved method use. However, whilst reading through the manuscript, I found some aspects that were not very clear for me.

Overall, the quality of the data, the attention to detail, make me believe the manuscript is ready for publication with some minor corrections.

Author reply:

Dear reviewer, thank you for your time, thoughtful questions, and genuine attitude toward the manuscript improvement. We tried our best to address all your concerns.

Questions:

- 1) At the Introduction authors mentioned the newly developed method allowed to measure the SABRE exchange parameters for other substrates including [1- ¹³C]pyruvate. It should be provided citation if so. Also, the manuscript is describing study of ¹⁵N-substrates and nothing regarding ¹³C-substrates or especially pyruvate, perhaps it would be better not to mention it here.

Author reply:

We guess you are referring to the last sentence “The newly developed method allowed us to measure the SABRE exchange parameters for other substrates including [1-¹³C]pyruvate.”

You are right, it is rather confusing as below we do not discuss it.

We already have preliminary data for the [1-¹³C]pyruvate exchange rates using SABRE, which will be submitted soon. However, to avoid confusion we removed this sentence from the Introduction and modified the Conclusions a bit.

Changes made to the manuscript:

New sentence in conclusion:

For example, recently, we measured *J*-coupling constants between IrHH protons and coordinated [1-¹³C]pyruvate using a modified INEPT sequence¹⁴. A slight modification of the sequence presented here will allow one to gain insight into the chemical kinetics of pyruvate exchange during SABRE.

Question:

2) Why for N3-MNZ has been used different conditions (high-pressure tube instead of low-pressure as for others, different time of catalyst activation)?

Author reply:

Thank you for the question. Yes, with the ambient pressure system, the polarization of the hydrides of the complexes after PASADENA was too low to transfer it to the nitrogen-15 nucleus and study the exchange rates. That's why we used the high-pressure system to overcome this problem.

Changes made to the manuscript:

In the second sub-section of the Results and Discussion section the following comment was added to address the question:

Note that experiments for all compounds were conducted at ambient pressure except for experiments with MNZ. This is because preliminary tests at ambient pressure with MNZ revealed the necessity for a long activation exceeding one hour and did not provide sufficient signal gain for reliable measurements. Using a higher pressure (6.6 bar), a common practice to increase activation rate and polarization yield⁵¹, the sample was activated within 40 minutes, and both phINEPT+ and ESOTHERIC provided close to 0.8% ¹⁵N polarization.

Question:

3) Ref.48 should be checked as it doesn't referred to the text.

Author reply:

Sorry, thanks for pointing it out. It was a mistake in referencing. We checked all instances of using ref 48.

The first instance:

“The efficiency of this approach, though, can be improved further if frequency-selective radiofrequency (RF) pulses are used instead of hard pulses to excite the protons originating from pH₂.⁴⁸ “

We see that it fits here as the paper mentions the importance of the frequency of selective pulses and the enhancement of polarization. Also, we could support the reference by adding two other references.

The second instance:

“connected to a high-pressure system operating at 6.6 bar (similar to the one used in Ref ⁴⁸).”

We agree with you that it is wrong here. We replaced it with the correct reference where the high pressure system was used.

The third instance:

“effective 4-spin system (two hydrides IrHH and two equatorial ¹⁵N nuclei),⁴⁸”

We deleted the reference as the meaning is already clear within the context of the paper.

Changes made to the manuscript:

First:

pulses are used instead of hard pulses to excite the protons originating from p H_2 .^{14,46,48}

Second:

connected to a high-pressure system operating at 6.6 bar (similar to the one used in Ref.⁵⁹).

Third:

effective 4-spin system (two hydrides IrHH and two equatorial ^{15}N nuclei),

Question:

4) What pressure has been used for other samples?

Author reply:

Thank you for your question, we referred to the section on Methods in **phINEPT+, ESOTHERIC, SABRE-INEPT, and SABRE-ESOTHERIC experiments**. All samples were attached to an ambient pressure bubbling system.

“The NMR tube was then attached to an ambient pressure bubbling system⁵⁸ and placed inside the spectrometer”

The backpressure here is set at ambient pressure. Note that bubbling of p H_2 creates overpressure of between 0.2 and 0.3 bar. Also, we referred to the reference in which this system was used and described in details.

Changes made to the manuscript:

Original text:

All experiments with hyperpolarization were carried out on a Bruker 400 MHz spectrometer using a 5 mm BBO probe. A 600 μ L sample was transferred to a low-pressure 5 mm NMR tube (507-TR-8, Wilmad Lab). The NMR tube was then attached to an ambient pressure bubbling system⁵⁸ and placed inside the spectrometer.

Modified text:

All experiments with hyperpolarization were carried out on a Bruker 400 MHz spectrometer using a 5 mm BBO probe. Typically, a 600 μ L sample was transferred to a screw-cap 5 mm NMR tube (507-TR-8, Wilmad Lab). The NMR tube was then attached to a low-pressure system, which enabled p H_2 bubbling through the solution using 0.2-0.3 bar overpressure with respect to the ambient pressure⁵⁸, and placed inside the spectrometer. First, we flushed the sample with p H_2 for 30 min to activate the catalyst. .

Question:

5) Did the pressure of parahydrogen bubbling make any difference for polarization level for different samples?

Author reply:

Yes, but we did not quantify it and tested only for MNZ. It is rather known effect and we refer also to our previous answer and text modification.

Changes made to the manuscript:

Results, second section:

Note that experiments for all compounds were conducted at ambient pressure except for experiments with MNZ. This is because preliminary tests at ambient pressure with MNZ revealed the necessity for a long activation exceeding one hour and did not provide sufficient signal gain for reliable measurements. Using a higher pressure (6.6 bar), a common practice to increase activation rate and polarization yield⁵¹, the sample was activated within 40 minutes, and both phINEPT+ and ESOTHERIC provided close to 0.8% ¹⁵N polarization.

Question:

6) Did the authors measure the sample volume after catalyst activation and after bubbling during experiments as well? It might be that after 30 - 40 min of bubbling, the methanol evaporates to give an optimised concentration which then becomes too concentrated and is affected detrimental to the polarization observed

Author reply:

Good point, we considered it but unfortunately did not describe it in the submitted versions of the manuscript.

Changes made to the manuscript:

After activation, we conducted experiments with the same sample for up to 5 hours. During the course of the experiment, the sample volume decreased by less than 10% thanks to the low temperatures used. Still, this leads to additional errors when evaluating the exchange rate constants. To avoid this, two solutions were proposed (not used here): adding a sample reservoir⁶⁴ that keeps the sample volume constant or saturation of dry pH₂ with methanol that significantly slows down evaporation^{29,65}. Note that the latter approach works thanks to the slow para-to-ortho conversion of H₂ in clean solvents and tubes⁶⁶.

Question:

7) At the Results and Discussion section authors again talking about “experimentally varied the two-time intervals τ_1 and τ_2 and measured the ¹⁵N signal of the bound ¹⁵N-labeled substrates” at different temperatures for N3-MNZ and other samples.

- If this aspect concerning N3-MNZ has already been presented somewhere, it should be referred to, if not, then an explanation will need to be made in the current manuscript.

Author reply:

If we understood correctly, the question concerns the reasons behind using different temperatures for MNZ and other substrates. Indeed, our rationale for using lower temperature for MNZ was not properly explained in the submitted manuscript. We modified the text, hopefully our rationale is now clear.

Changes made to the manuscript:

Original text:

Then, we experimentally varied the two time intervals τ_1 and τ_2 (**Figure 2A, B**) and measured the ^{15}N signal of the bound ^{15}N -labeled substrates at 288 K for ^{15}N -Py, ^{15}N -NAM, ^{15}N -4AP, and ^{15}N -ACN and 267 K for $^{15}\text{N}_3$ -MNZ (**Figure 3**). We used a 1 ms time grid to assess the optimal conditions that required from 25 to 81 experiments per map with 453 hyperpolarization experiments in total.

New text:

For this, we experimentally varied the two intervals τ_1 and τ_2 (**Figure 2A, B**) and measured the ^{15}N signal of the bound ^{15}N -labeled substrates at 288 K for ^{15}N -Py, ^{15}N -NAM, ^{15}N -4AP, and ^{15}N -ACN and 267 K for $^{15}\text{N}_3$ -MNZ (**Figure 3**). At 288 K, $^{15}\text{N}_3$ -MNZ did not provide high polarization values, as will be evident below, due to the rapid dissociation rate—one of the fastest among tested substrates. Therefore, we cooled the sample down to 267 K where the rate sufficiently slowed down, enabling polarization of 0.8% at optimal SOT parameters. We used a 1 ms time grid to assess the optimal conditions required from 25 to 81 experiments per map, with 453 hyperpolarization experiments in total. Found this way, optimal SOT parameters were used in all other experiments, including different temperatures for $^{15}\text{N}_3$ -MNZ and other substrates: this is justifiable because although exchange rates are changing significantly as a function of temperature, the spin-spin interactions—driving force for SOT—do not change significantly in the studied range.

Question:

8) The dissociation rates for the five substrates under study were significantly different. “ ^{15}N -ACN and $^{15}\text{N}_3$ -MNZ exchanged significantly faster than the pyridine derivatives ^{15}N -Py, ^{15}N -4AP, and ^{15}N -NAM”. Do you have any explanation for it? It would be useful to provide them here.

Author reply:

Thank you. We have a plausible explanation but ideally it should be tested on a larger pool of tracers to be confirmed.

Changes made to the manuscript:

Added the following text in Results and Discussion: Exchange rates sub-section.

It has been predicted that, for pyruvate derivatives, in the absence of steric effects, the lower the pKa of the conjugate acid, the stronger the ligand binding⁶⁷. ACN and MNZ do not belong to the six-membered heterocycle family; the higher charge density of six-membered heterocycles leads to stronger interactions between charged entities (Ir^+ and the ring) and slows dissociation rates. In addition, MNZ has additional steric hindrances. A rationale for the rapid loss of ^{15}N -ACN (the smallest molecule in the group) needs further evaluation, although these data are consistent with other studies, as shown in Ref⁶⁸. To confirm the influence of the reagent structure, a larger pool of substrates, including five-membered rings with 1-3 nitrogens, should be tested.

Question:

9) In the Conclusion authors are saying,

“Moreover, we envision that even at a natural abundance of ^{15}N nuclei with high enough polarization levels exceeding 50%, ^{66}Ir -complexes' lifetime can be studied using the chemical models presented here, even at low field benchtop NMR spectrometers.” Perhaps it might make sense to test at least natural pyridine to confirm this, and maybe also test the current system on benchtop NMR.

Author reply:

Thank you for the inspiration. We did only experiments with n.a. pyridine, and it worked. A new section with these results was added. The tests on the benchtop spectrometer are planned in the future, but we feel that they are out of scope of this article. Also we added additional changes to the last part of the Conclusions.

Changes made to the manuscript:

We added a completely new sub-section in the Results and Discussion section:

Hyperpolarization of naturally abundant (n.a.) pyridine. To assess the hyperpolarization efficiency, we attempted to hyperpolarize unlabelled 400 mM pyridine (the natural abundance of ^{15}N isotope is about 0.4%) with 4 mM IrIMes catalyst. We experimentally transferred polarization to equatorial pyridine using both phINEPT+ and ESOTHERIC pulse sequences (Figure 7A) and measured exchange rates (Figure S11, SI) that gave deviation of up to 20% compared to values measured with labeled ^{15}N -Py (Table S3). The maximum achievable polarization for equatorially bound pyridine was ~7.2% using phINEPT+ and ~17.5% using ESOTHERIC: ESOTHERIC provided twice the signal than phINEPT+. This is because the system was reduced effectively from a four-spin system consisting of two IrHH protons and two ^{15}N nuclei of two equatorial substrates to the system with only one ^{15}N nucleus in the equatorial plane. In this case, ESOTHERIC performance is optimal, while phINEPT+, at best, can give 50% of ESOTHERIC polarization values. To confirm our measurements, we simulated SOT performance for three and four spin systems without chemical exchange to estimate performance in the case of n.a. Py (Figure 7C) and ^{15}N -Py (Figure 7D). The data qualitatively agree with experimental observation.

And in Conclusions, the last part was modified:

Through this methodology, it would be possible to evaluate the wide-ranging number of transition metal⁷⁴ and metal-free⁷⁵ species detected through PHIP. Transfer to ^{31}P in the context of phosphine, and ^{13}C in the context of CO represent two of the most important ligands in catalysis. Furthermore, as ^{31}P is essentially 100% abundant, labeling is not needed, and it may be possible to extend these approaches to normal NMR without hyperpolarisation. In the context of ^{13}C , the 1% natural abundance suggests hyperpolarisation will be necessary, but using ^{13}CO is also relatively simple. For example, recently, we measured J -coupling constants between IrHH protons and coordinated [1- ^{13}C]pyruvate using a modified INEPT sequence¹⁴. A slight modification of the sequence presented here will allow one to gain insight into the chemical kinetics of pyruvate exchange during SABRE. This approach is also not limited to RF SOT-driven hyperpolarization. With demonstrated ^{15}N polarization values achieved in ultra-low field experiments⁷⁷, kinetics measurements even on low-field benchtop NMR systems should be possible. As the accuracy of this approach has been shown to be high, the results will be significant.

And in the Abstract the following sentence was added:

The presented approach can be successfully applied not only to isotopically enriched substrates but also to compounds with natural abundance of the to-be-hyperpolarized heteronuclei.

Changes made to the Supporting Information:

Figure S11 and the corresponding section,

N.a. pyridine exchange

We performed SABRE-ESOTHERIC measurements using pyridine with the natural abundance of ^{15}N nuclei. The sample was prepared by mixing 400 mM of pyridine with 4 mM of $[\text{Ir-d}_{22}]$ in 0.6 mL of methanol- d_4 . First, we experimentally estimated the optimum parameters for phINEPT+ and ESOTHERIC for polarization transfer from pH_2 -derived hydrides to the equatorially bound pyridine on the SABRE catalyst. We varied the two time intervals τ_1 and τ_2 (**Figure 7A**) and measured the ^{15}N signal of the bound ^{15}N natural abundance pyridine at 280 K. The SABRE-ESOTHERIC kinetics for n.a. pyridine are presented in **Figure S11**.

Figure S11. Dependences of the SABRE-ESOTHERIC-enhanced ^{15}N NMR signals of n.a. ePy (blue) and n.a. fPy (red) on the inter-pulse delay τ_e with $\tau_1 = 9$ ms and $\tau_2 = 6$ ms at the nominal temperatures of 280 K (A), 283 K (B), 288 K (C), and 293 K (D). The curves are fitted using biexponential global fit. Estimated exchange rate constants are given in **Table S3**.

Table S3 was expanded with additional data on n.a. pyridine exchange.

Reviewer #2:

Recommendation:

I recommend the publication of the manuscript following revisions to the following points:

Comments:

Overall, this manuscript elegantly explores spin physics with broad implications in biochemistry and medical imaging.

Author reply:

Dear reviewer, thank you for your time, thoughtful questions, and genuine attitude in improving the manuscript. We tried our best to address all your concerns.

Question:

1) The description of IzSz as purely singlet order may not hold if the two spins exhibit similar frequencies. The authors should clarify this point, as the protons in the study appear to have similar chemical shifts.

Author reply:

In principle, we did not use such a description explicitly, or do we oversee it somewhere? If so, please point it out. At the same time, the point is legit, and one could expect to have a singlet state as an initial spin state of the system consisting of two chemically equivalent protons, which is not the case here. The explanation and references were added to the text.

Changes made to the manuscript:

Added two sentences in the results and discussion:

Although here only symmetric spin systems were studied, resulting in chemically equivalent two hydrides IrHH, the fact that these nuclei are strongly magnetically nonequivalent results in averaging singlet state, leaving imbalanced observable antiphase $\hat{I}_z\hat{S}_z$ spin order^{60,61}. The discussed pulse sequences are designed to transfer this type of polarization to heteronuclei.

Question:

2) The ESOTHERIC sequence involves a longer pulse train than phINEPT. Does this affect the measurement of fast exchange rates?

Author reply:

Indeed, the ESOTHERIC sequence involves a longer pulse train than the phINEPTplus sequence; however, even at high temperatures with its fast exchange rates, the measurements were not affected, and ESOTHERIC was not worse.

Moreover, we introduced a total time of SOTs and now the reader can see that in fact always ESOTHERIC is not longer than phINEPT+ because the optimized inter-pulse delays for the former sequence turned out to be shorter.

Changes made to the manuscript:

t_{tot} time was introduced in Figures 2 and 3, and one comment was added in the results:

Although the ESOTHERIC sequence has more RF-pulses and time intervals, the total duration of the phINEPT+ sequence is not shorter than that of ESOTHERIC, when both are at optimal settings for all studied here systems (**Figure 3**): an additional positive feature of ESOTHERIC sequence.

Question:

3) There seems to be more phase distortion in the ESOTHERIC case compared to phINEPT, as shown in Figure 3. Can the authors clarify the origin of this distortion?

Author reply:

We appreciate your interest in this effect and added the discussion in the text and the SI.

Changes made to the Supporting information:

Section “Phase distortion in the ESOTHERIC SOT” was added in the SI

Phase distortion in the ESOTHERIC SOT sequence

The spectra plotted in **Figure 3** of the manuscript show NMR line distortions. These distortions are coming from zero and higher-order quantum coherences and the fact that the spectra are measured directly after ESOTHERIC or phINEPT+ SOT sequences without any filters. To confirm that, we compared SABRE-ESOTHERIC NMR spectra measured with pulsed field gradient (PFG, filter of quantum coherences above one) during τ_e interval, with pulse field gradient and a zero-quantum (ZQ) filter and without any of this (**Figure S1**). Both PFG and ZQ filters contribute to the improvement of the line shape. The kinetic measurements during the study used only PFG. As exemplified here, the contribution of the ZQ filter changes the amplitude only by about 6% but, at the same time, does not allow for the measurement of exchange at times below 10 ms due to the ZQ filter.

Figure S1. ¹⁵N spectra of equatorial pyridine hyperpolarized with SABRE-ESOTHERIC. PFG and ZQ filters were added between the last two pulses (see **Figure 2C**): spectrum without any filters (red), spectrum with 2 ms of PFG (31% of SMSQ10.100, green), and spectrum with the same PFG and

10 ms of ZQ filter (blue). Integrals were measured to check the difference between these measurements, which showed an 8% higher signal using only PFG and 14% higher signal when using both PFG and ZQ filters.

And in Results and Discussion section in the manuscript, we added the following paragraph:

The spectral lines of hyperpolarized molecules measured immediately after phINEPT+ and ESOTHERIC have distorted line shapes due to polarization of zero and higher-order quantum coherences, which are populated due to numerous spin-spin interactions in the system. These coherences can be suppressed by adding a corresponding quantum filter⁶³. In the following experiments with SABRE-INEPT and SABRE-ESOTHERIC (SOT scheme in **Figure 2C**), we used pulsed field gradient to filter quantum coherences of higher order, which was found to be sufficient to provide distortionless lines (**Figure S1, SI**).

Question:

- 4) The phase cycling schemes used in ESOTHERIC and phINEPT should be described in greater detail.

Author reply:

We did not use any phase cycling. To remove the higher order quantum coherences we used pulsed field gradient as discussed in the text and commented above.

Reviewer #3:

The manuscript covers a modification to the SABRE amplification of NMR signals using an experimental spin order transfer sequence where readout occurs at the ^{15}N nuclei by direct interaction with the catalyst. Using this method, ligand exchange models can be fitted giving dissociation rate constants enabling the study of the kinetics. This information can be critical for a thorough understanding of catalytic behaviour and, as such, this technique has the potential to be very useful in the development and understanding of new catalytic systems.

I look forward to seeing the results from others using this technique in due course.

Author reply:

Dear reviewer, We thank you for your appreciation of our research.

Question:

- 1) This referee is not an expert in NMR, however, it would seem that there are limitations to this method. The authors comment towards the end that their approach is "not limited to the RF SOT-driven hyperpolarisation of ^{15}N labelled compounds". However, it would seem to be at least appropriate to mention the applicability to other metal nuclei than Ir, in concept even if the authors have not studied this themselves.

Author reply:

Thank you for pointing that out. We extended the section in conclusion to account for that.

Changes made to the manuscript:

Modified text in conclusions:

Through this methodology, it would be possible to evaluate the wide-ranging number of transition metal⁷⁴ and metal-free⁷⁵ species detected through PHIP. Transfer to ^{31}P in the context of phosphine, and ^{13}C in the context of CO represent two of the most important ligands in catalysis. Furthermore, as ^{31}P is essentially 100% abundant, labeling is not needed, and it may be possible to extend these approaches to normal NMR without hyperpolarisation. In the context of ^{13}C , the 1% natural abundance suggests hyperpolarisation will be necessary, but using ^{13}CO is also relatively simple. For example, recently, we measured J -coupling constants between IrHH protons and coordinated $[1-^{13}\text{C}]$ pyruvate using a modified INEPT sequence¹⁴. A slight modification of the sequence presented here will allow one to gain insight into the chemical kinetics of pyruvate exchange during SABRE. This approach is also not limited to RF SOT-driven hyperpolarization. With demonstrated ^{15}N polarization values achieved in ultra-low field experiments⁷⁷, kinetics measurements even on low-field benchtop NMR systems should be possible. As the accuracy of this approach has been shown to be high, the results will be significant.